# Does Diversity in Top Management Teams Contribute to Organizational Performance? The Response of the IBEX 35 Companies

**Mercedes Rodríguez-Fernández \*** **, Ana I. Gaspar-González and Eva M. Sánchez-Teba**

Department of Economics and Business Administration, Campus El Ejido s/n 29071, University of Málaga, 29016 Málaga, Spain; aigaspar@uma.es (A.I.G.-G.); emsanchezteba@uma.es (E.M.S.-T.)
\* Correspondence: mmrodriguez@uma.es

**Abstract:** This study contributes to the dissemination of the theoretical and empirical knowledge on the Upper Echelons Theory, considering training and demographic diversity in Top Management Teams (TMTs) as a unique feature of companies, in our case, the IBEX 35 companies. Based on the results, we can confirm that the inclusion of women in management teams positively influences the sales of a company and contributes to increasing financial results. Age and knowledge of two or more languages are important factors in achieving an increase in financial performance. From the point of view of business practice, the results obtained are useful for increasing knowledge of which TMT characteristics are valid, which allows for better results and the establishment of responsible organizational policies that promote the inclusion of gender diversity in TMTs. In addition, the results of this study indicate that the incorporation of members of other non-Spanish nationalities would constitute a distinctive feature of a company and would enrich it not only financially, but also culturally.

**Keywords:** diversity; top management team; upper echelons theory; organizational performance; IBEX 35 companies; net sales; EBITDA

## 1. Introduction

Diversity is the range of characteristics that can be associated with an organization, individual, or system. This concept refers to the "human characteristics that make people different from each other" (Gomez-Mejia et al. 2007). Gómez-Mejía et al. argue that the range of characteristics includes both those characteristics that are not controllable by the individual and those over which a certain amount of control can be exerted. They distinguish between a typology of characteristics, called the first category, which are biologically determined, like age, race and gender, and the second category includes the characteristics that people can adopt, abandon, or change throughout their lives by choice and intentional decisions. These include work experience, marital status, education, etc.

However, Kossek and Lobel (1996) provide a more restrictive perspective on the concept, considering diversity only in terms of race, gender, and ethnicity, that is, not including the heterogeneity arising from the characteristics of the individual that are adopted intentionally in the concept of diversity. Nevertheless, there are many other authors (Carr 1993; Carnevale and Stone 1995; Norton and Fox 1997) who do include a greater number of characteristics from both categories (controllable and non-controllable).

In this study, we analyze those human characteristics that make the people in an organization different and that could ultimately influence the strategic decisions made by a Top Management Team (TMT), which, in turn, affect organizational outcomes.

Throughout the literature on Human Resources management, it is possible to observe how these variables relating to diversity, considered in TMTs, can influence HR management processes and the making of business decisions (Nicholson and Kiel 2007; Dalton and Dalton 2011).

There are many opinions on how to explain diversity, ranging from the ideas that emerged in the 1970s and 1980s, which sparked the development of a more diverse workforce and which led to an appreciation of diversity among companies, to the opinions of authors who postulate that there is a positive relationship between diversity and organizational performance (Barney 1991).

Our analysis, accepting the existence of different typologies of diversity, i.e., those that include biologically determined characteristics and those voluntarily adopted, proposes that all these characteristics can lead to TMTs forming a series of values that influence decision-making processes and produce a positive or negative result in terms of organizational performance.

In the context defined above, we will focus on the Upper Echelon Perspective Theory (Hambrick and Mason 1984), which considers the qualities that influence top management decision-making and, consequently, organizational performance, while also determining if that diversity is a strategic instrument for organizational performance. As such, this theory is the theoretical framework within which the target population of our research is analyzed: The IBEX 35 companies' TMTs.

The work is structured as follows: First, we establish the conceptual framework and the starting hypotheses, then, we continue with the methodology and the analysis of the results, and finally, we draw the conclusions.

## 2. Conceptual Framework

In recent decades, interest in the relationship between the governing body of organizations and decision-making has grown. Hambrick and Mason (1984) were precursors in the study of this topic, arguing that the strategic decisions of governing teams or those who direct organizations, have great behavioral content since these decisions reflect the philosophy and particularities of those who make them. This idea formed the axis of the "Top Management Theory".

The focus of the Upper Echelons or "High Command Theory" is not centered exclusively on the CEO of an organization, but rather on the entire team of managers, i.e., the Top Management Team. The theory argues that organizational outcomes are linked to the personal characteristics of the members with a greater amount of power within an organization.

The theory postulates that organizational outcomes are strongly influenced by the decisions that senior executives make, which, in turn, are affected by managers' characteristics. One of the theory's fundamental aspects is that it claims that demographic characteristics, such as a management team's age, gender, or nationality, together with those responsible for making decisions in an organization, have an important influence on a company's performance.

Authors, such as Knight et al. (1999), point out that in the Upper Echelon Theory, demographic measures are linked to the "psychological and cognitive elements of the executive orientation".

This approach is based on the idea that an organization's top management is responsible for what happens in it, and therefore, the analysis of the personal characteristics of these managers will condition the organization's economic results, growth, performance, or innovation (Goll and Rasheed 2005; Hambrick 2007).

Based on the research carried out by Hambrick and Mason (1984), in this work, we will analyze a sample of companies to determine if the personal characteristics of the Top Management Teams influence their values, and if, as a consequence, they affect the organization's results.

According to the Upper Echelon Theory, the characteristics of a TMT, such as its demographic and training diversity, and its ability and capacity to take risks that lead to better corporate results, have been studied in large- and medium-sized companies (Ling and Kellermanns 2010; Minichilli et al. 2010). These senior management characteristics have been considered to be influencing variables of a company's results (Michel and Hambrick 1992; West and Schwenk 1996; Keck 1997).

*Statement of Hypotheses*

Firstly, the general aim of this article is to determine the influence exerted by the diversity of the IBEX 35 firms' management teams on company performance. Together with this purpose, and based on the theories and works analyzed, considering that Top Management Team members influence the development of their organizations, we propose the following hypotheses that underlie two more specific objectives:

**Specific Objective 1 (SO1):** *Determine the influence exerted by the demographic and formative diversity of the IBEX 35 companies' management teams on net sales.*

**Hypothesis 1a (H1a).** *The inclusion of women in the IBEX 35 companies' management teams positively influences net sales.*

In this first hypothesis, it is considered that a company's sales will depend to a great extent on the gender of the TMT. In this line, authors, like Hambrick and Finkelstein (1987) or Li and Tang (2010), aimed to determine how much female representation matters in relation to the performance of a company.

**Hypothesis 1b (H1b).** *Net sales of the IBEX 35 listed companies increase when members of the management teams are older.*

In this hypothesis, a better sales result is linked with the firms' Top Management Teams falling within an older age bracket. Authors, such as Zhang et al. (2017), point out that heterogeneity of age among the members of a TMT can significantly and positively affect business performance. It is, therefore, best to maintain the age level and continue to introduce new members when building a TMT.

**Hypothesis 1c (H1c).** *The inclusion of various nationalities in the management teams of the IBEX 35 companies favors an increase in business net sales.*

A positive relationship is established between the increase in company sales and the heterogeneity of nationalities of the TMT of an organization.

All types of diversity have been studied by many authors who, following the Upper Echelon approach, found that diversity can provide benefits. This prompted Mitchell et al. (2016) to analyze, among other things, national diversity as a variable determinant of when and how upper echelons' decisions generate positive or negative effects.

**Hypothesis 1d (H1d).** *The training of members of the IBEX 35 companies' management teams contributes to an increase in company sales.*

Our approach establishes a relationship between the complementary training of a management team, the training received to increase skills, which is studied within a company, the appropriate training for a position and the net sales of an organization. Along with diversity in gender, diversity in educational level stands as a fundamental variable in the development of new ideas and could, therefore, contribute to success in the results of an organization (Van der Vegt and Janssen 2003).

**Specific Objective 2 (SO2):** *To determine the influence exerted by the demographic and formative diversity of the IBEX 35 companies' management teams on profit, measured in terms of EBITDA.*

**Hypothesis 2a (H2a).** *The inclusion of female members in management teams has a positive effect on the EBITDA of the IBEX 35 companies.*

The effect of gender on business performance has been studied in many developed countries (Westhead 2003; Khalife and Chalouhi 2013). The increase in the role of women in the decision-making of the company reveals a significant increase in organizational results (Watson 2012; Lee and Marvel 2014).

**Hypothesis 2b (H2b).** *The final EBITDA of the IBEX 35 listed companies increases when the members of the management teams are older.*

Mayr (2011) considers that the average age of the Top Management Team determines their performance in the company since an increase in the management teams' average age is translated into an accumulation of experience and knowledge (Bantel and Jackson 1989). In line with Hambrick and Mason (1984), age forms managers' cognitive basis and values.

**Hypothesis 2c (H2c).** *The inclusion of various nationalities in the management teams of the IBEX 35 companies favors an increase in the final EBITDA.*

The cultural characteristics of companies with teams of transnational managers of different nationalities allow for a better understanding of the international market (Bouncken and Winkler 2010) and consequently influence the benefits of the organization.

**Hypothesis 2d (H2d).** *The training of members of the IBEX 35 companies' management teams contributes to an increase in the companies' EBITDA.*

The last hypothesis raises a positive relationship between the management team training of the IBEX 35 companies and an increase in their EBITDA.

## 3. Methodology

The general objective of this research is to empirically determine the influence exerted by the diversity of the IBEX 35 companies' Management Teams on the companies' results. In addition, two specific objectives are established: SO1 and SO2, which specifically focus on analyzing the influence exerted by two types of diversity in these listed companies: Demographic and educational diversity. To achieve this goal, and following a theoretical review, several hypotheses are posed, and a methodological proposal, which allows for a comparison of these hypotheses, is introduced.

The set of hypotheses in the study links with the general objective based on the theory explained above. Each of the formulated hypotheses aims to satisfy the specific objectives outlined using multiple linear regression and the Ordinary Least Squares (OLS) estimators. This method allows for the quantification of the relationship among the variables, obtaining an approximation of the magnitude of their influence—in our case, on net sales increase and final corporate EBITDA. The method also allows for a comparison of all the hypotheses and for the specific objectives proposed to be met. The analysis method is valid for estimating the influence of the diversity of the members of corporate management on business performance (Angrist and Pischke 2009) and has been used by other authors (Knežević et al. 2017) with the same aim.

### 3.1. Sample and Data

For the analysis, the data extracted from the SABI (Iberian Balance Sheet Analysis System) database were used for the dependent variables, NET SALES and EBITDA. For the rest of the variables, the necessary data of the Annual Reports, published by each IBEX company, have been reviewed and selected.

The data that make up the sample were collected in 2016. The entire banking sector present in the IBEX of that year was excluded from the sample, which is consistent with previous studies.

Therefore, once this sector had been excluded, complete data were obtained for the remaining IBEX companies. These include all the board members and directors of the 28 IBEX 35 companies

(the remaining seven correspond to the banking sector), so there can be no sampling error. The sample population size corresponds to N = 419[1] people from the manager, director, and dual role positions.

### 3.2. Variables

Table 1 shows a statistical description (number of cases, minimum, maximum, mean, and standard deviation) of the variables included in the regression. In the first place, there is the dependent variable, NET SALES, referring to the annual sales obtained by each IBEX company, which is continuous quantitative and demonstrates the commercial capacity of each firm. The dependent variable, EBITDA (quantitative continuous), the gross operating profit of each IBEX 35 company, calculated before deducting financial expenses and taxes, since we estimate that it is each company's full result, is shown below.

The rest of the variables used in the analysis, shown in Table 1, are introduced using a theoretical basis, since the upper echelon theory, explained above, clarifies the idea that the characteristics of the management team of the firm can be reflected in the results of the organization (Hambrick and Mason 1984). Thus, we hold that both demographic and formative diversity variables can influence business performance, and they are represented, in this analysis, by the dependent variables, mentioned above.

The demographic variables include GENDER, AGE, and NATIONALITY, and the rest of the variables correspond to the training and work experience of all members of the management teams.

**Table 1.** Variable descriptive statistics included in the regression.

| | N | MINIMUM | MAXIMUM | MEAN | DEV. TYP. |
|---|---|---|---|---|---|
| NET SALES (thousands) | 419 | 8166.00 | 38,372,521.00 | 5,988,817.71 | 9,494,382.64 |
| EBITD (thousands) | 419 | −100,7391.00 | 5,324,107.00 | 922,335.9189 | 1,440,716.210 |
| GENDER (Men) | 419 | 0 | 1 | 0.83 | 0.373 |
| AGE<br>AGE 1 (Between 31–45) ** | 419 | 0 | 1 | 0.03 | 0.180 |
| AGE 2 (Between 46–60) ** | 419 | 0 | 1 | 0.29 | 0.456 |
| AGE 3 (Between 61–70) ** | 419 | 0 | 1 | 0.25 | 0.434 |
| AGE 4 (+70) ** | 419 | 0 | 1 | 0.13 | 0.333 |
| NATIONALITY (Spanish) | 407 | 0 | 1 | 0.85 | 0.362 |
| TRAINING LEVEL<br>UNIVERSITY EDUCATION | 392 | 0 | 1 | 0.99 | 0.112 |
| DOCTORATE | 392 | 0 | 1 | 0.14 | 0.342 |
| DOUBLE DEGREE | 355 | 0 | 1 | 0.36 | 0.482 |
| ECONOMIC TRAINING | 385 | 0 | 1 | 0.42 | 0.494 |
| LEGAL TRAINING | 385 | 0 | 1 | 0.27 | 0.443 |
| TECHNICAL FORMATION | 385 | 0 | 1 | 0.24 | 0.425 |
| MASTER'S DEGREE | 301 | 0 | 1 | 0.42 | 0.494 |
| LANGUAGES<br>NATIVE LANGUAGE + 2 O + | 327 | 0 | 1 | 0.33 | 0.471 |
| OTHER TRAINING<br>TRAINING SUITABLE FOR THE POSTI * | 375 | 0 | 1 | 0.83 | 0.374 |
| FURTHER TRAINING | 265 | 0 | 1 | 0.63 | 0.483 |
| EXPERIENCE<br>EXPERIENCE OTHER SECTORS | 402 | 0 | 1 | 0.19 | 0.396 |
| INTERNATIONAL EXPERIENCE | 357 | 0 | 1 | 0.76 | 0.428 |

\* This variable refers to an employee that has a direct and specific relationship with the post at the present time, while further training corresponds to non-specific, transversal, and general training. \*\* Fictional variables are recoded (dichotomous) from the quantitative continuous originals. Own Elaboration. Data Source: SABI database. IBEX companies' annual reports.

---

[1] From a population of 419 members studied, 324 are managers, and 174 are directors. The mismatch of 79 people with the 419 that make up the sample size corresponds to those who have dual roles (directors and managers who have worked in the same company for the same amount of time). Therefore, 245 are solely managers, and 95 are solely directors.

Regarding the construction of the variables, we found AGE, before quantitative continuous, for ages of 31 to 95 years, and this has been recorded and transformed into four dichotomous fictitious variables: AGE 1, AGE 2, AGE 3, and AGE 4 (with values of 1 = Yes and 0 = No), comprising the ages listed in Table 2.

The remaining independent variables related to training and experience, together with the variable, GENDER (whose population, N = 419, consists of 16.7% women and 83.3% men), are all qualitative dichotomous. The construction of all the variables is explained in Table 2.

**Table 2.** Variables construction.

| Dependents | | | |
|---|---|---|---|
| 1.NSL | NET SALES (thousands) | NUMERIC | |
| 2.EBI | EBITDA (thousands) | NUMERIC | |
| **Independents** | | | |
| 1.GENDER1 | GENDER | {0, WOMAN} | {1, MAN} |
| 2.AGE1 | 31–45 | {0, NO} | {1, YES} |
| 3.AGE2 | 46–60 | {0, NO} | {1, YES} |
| 4.AGE3 | 61–70 | {0, NO} | {1, YES} |
| 5.AGE4 | +70 | {0, NO} | {1, YES} |
| 6.NAT1 | NATIONALITY | {0, REST N} | {1, SPANISH} |
| 7.TRAIN1 | UNIVERSITY TRAINING | {0, NO} | {1, YES} |
| 8.DOCT1 | DOCTORATE | {0, NO} | {1, YES} |
| 9.DOBDEG1 | DOUBLE DEGREE | {0, NO} | {1, YES} |
| 10.ECONTRAIN1 | ECONOMIC TRAINING | {0, NO} | {1, YES} |
| 11.LEGTRAIN1 | LEGAL TRAINING | {0, NO} | {1, YES} |
| 12.TECTRAIN1 | TECHNICAL TRAINING | {0, NO} | {1, YES} |
| 13.MASTER1 | MASTER TRAINING | {0, NO} | {1, YES} |
| 14.LANGUAGE3 | NATIVE LANGUAGE + 2 O + | {0, NO} | {1, YES} |
| 15.TPOST1 | TRAINING SUITABLE POST | {0, NO} | {1, YES} |
| 16.COMPLT1 | COMPLEM. TRAINING | {0, NO} | {1, YES} |
| 17.EXPOTR1 | EXPERIENCE OTHER SECTORS | {0, NO} | {1, YES} |
| 18.INTEXP1 | INTERNATIONAL EXPERIENCE | {0, NO} | {1, YES} |

Own Elaboration Data source: SABI database. Annual reports of IBEX companies.

## 4. Analysis of Results

The composition of the diversity of team members is another important characteristic of the behavior of organizations, which differentiates the companies. Some researchers claim that gender diversity can be positive for a company to the point that the presence of women on boards can help a company to achieve an improvement in their decision-making processes and enhance their performance (Coffey and Wang 1998; Rose 2007). The data analyzed offer us the possibility of detailing the distribution of the TMT in terms of gender diversity and the position held (Figure 1).

Thus, it can be seen that those who solely hold the position of manager (245 members) are 79.59% male and only 20.41% female. The number of members who have the sole position of director is much lower (95), and the female percentage of directors is also lower, constituting only 11.58% of the population, while their male counterpart accounts for 88.42%. This is repeated again in the case of those who have dual roles as directors and managers, where 11.39% are female, and 88.61% are male. Therefore, for all three positions, females are underrepresented. This imbalance between both sexes has been previously studied in organizations (Wharton and Baron 1987) and companies present in the Stock Market (Rodríguez-Domínguez et al. 2012).

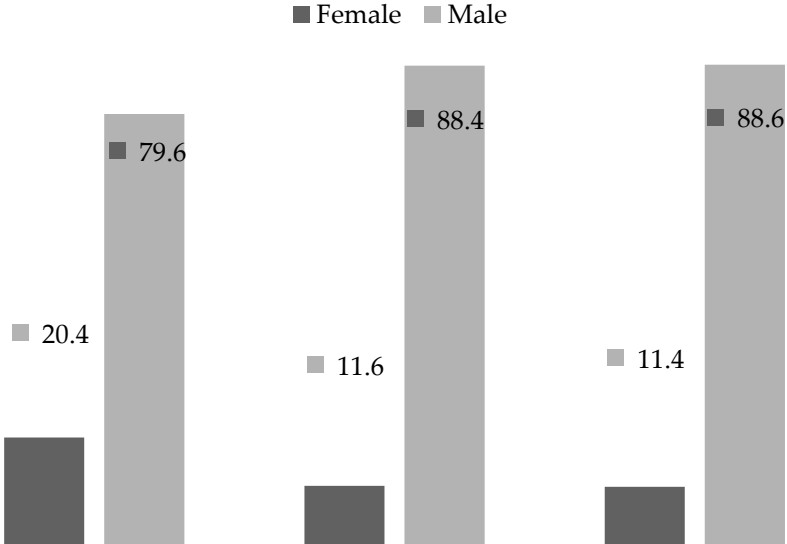

**Figure 1.** Distribution of the IBEX 35 companies' management teams by position and gender. Source: SABI database. IBEX companies' annual reports.

When distribution by gender and age group is analyzed, the same pattern is repeated regarding the low percentage of women in each position (Figure 2), but considering the variable, AGE, and different age groups, we can see the evolution experienced by women over time in each of the positions, when compared with males and with respect to the total team members.

The highest male percentages are found in the 61–70 and 46–60 age groups, while in general, the highest female percentages are found in the younger age group (31–45 and 46–60), with the exception of managers, which show an increase in the 61–70 age group, compared to the small percentage of female managers aged 31–45. In the +70-age group, there are no managers, which shows that in the older generations, the presence of women is non-existent and that they are incorporated later, once they have had more training and, consequently, greater participation in the world of work. Thus, the highest percentage of female managers (within their sex) are in the 46–60 age group (11.23 %). With regard to male directors, the highest percentage is found in the 61–70 age group (31.55%), and the lowest percentage (1.60%) corresponds to the youngest directors.

Male directors in the 46–60 years group stand out as the highest percentage of the entire distribution (47.37%), and the lowest percentage (same gender) is found in the youngest managers (10.53%). In contrast, the female figure in managerial positions is almost non-existent. In fact, it is null in the older age groups (61–70 and +70) and shows a low representation in the younger age group (2.63%). Moreover, although the middle-aged group, 46–60 years, has the highest proportion (5.26%) of females, it is insignificant when compared with the male percentage in the same age bracket, which confirms what was previously explained in relation to the manager position.

Finally, the dual role of director and manager follows the same pattern. There is a greater proportion of dual roles among men (the highest being 37.14% in the 61–70 years group), compared to the negligible percentage of women (the highest being 5.71% in the 46–60 years group). Again, the inexistence of females in the older age group is repeated.

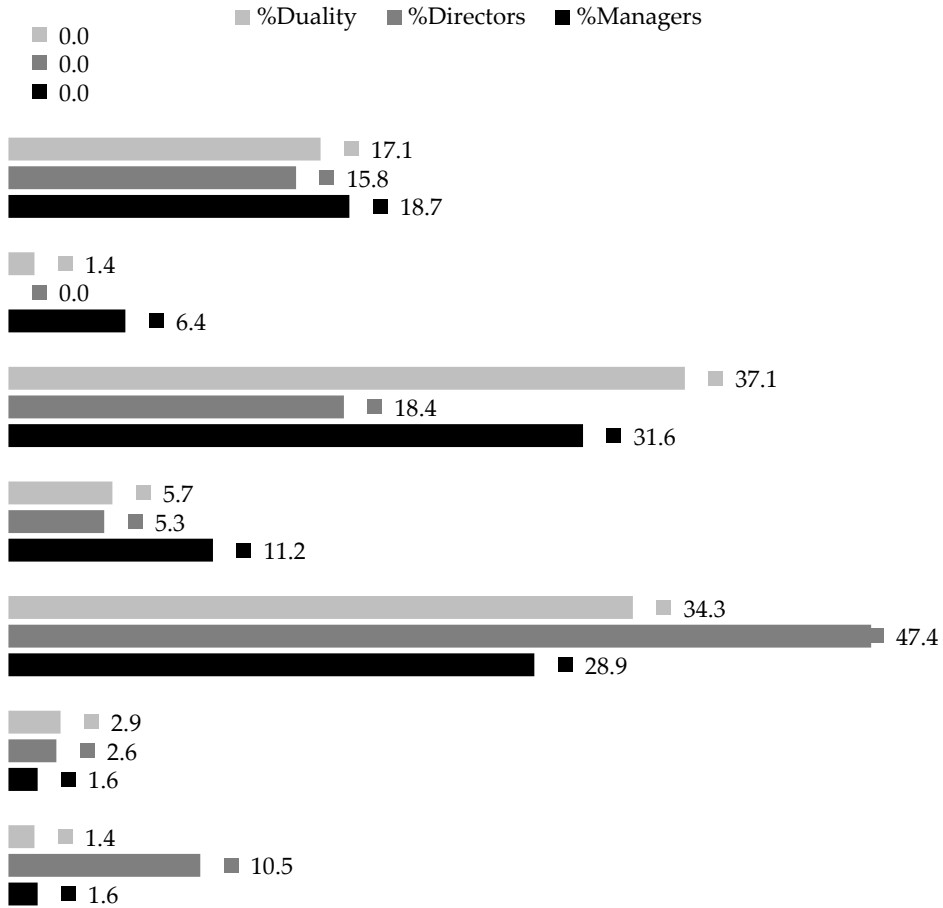

**Figure 2.** Distribution of Management Teams in the IBEX 35 companies by age group, position, and gender. Source: SABI database. IBEX companies' annual reports.

A good level of education among corporate management team members, regardless of their sex, is a characteristic that, combined with their skills, is favorable for the development of processes and decisions within a corporation, which produces good results for the firm (Norburn 1986; Gabrielsson and Winlund 2000; Rodríguez-Domínguez et al. 2012). Thus, if we observe the educational diversity of the IBEX 35 companies' TMT members (Figure 3), we find some changes with respect to the distributions analyzed above. As such, we can see that there is a higher percentage of female TMT members with Master's degrees than male members in dual roles and manager positions. This is repeated for those with doctorates holding a manager's position, which shows that females must have a higher education than their male colleagues if they want to advance in their profession, and this pattern is repeated at higher levels.

The type of training for the majority of management team members is economic, followed by legal and, to a lesser extent, technical. In these three training types, the male percentage is higher than the female percentage in the case of directors, although there is very little difference. However, female directors outperform male members in terms of their percentage (3.46 points) in the case of legal training and dual roles. The female proportion of executives and managers with training in two or more languages are also superior in number.

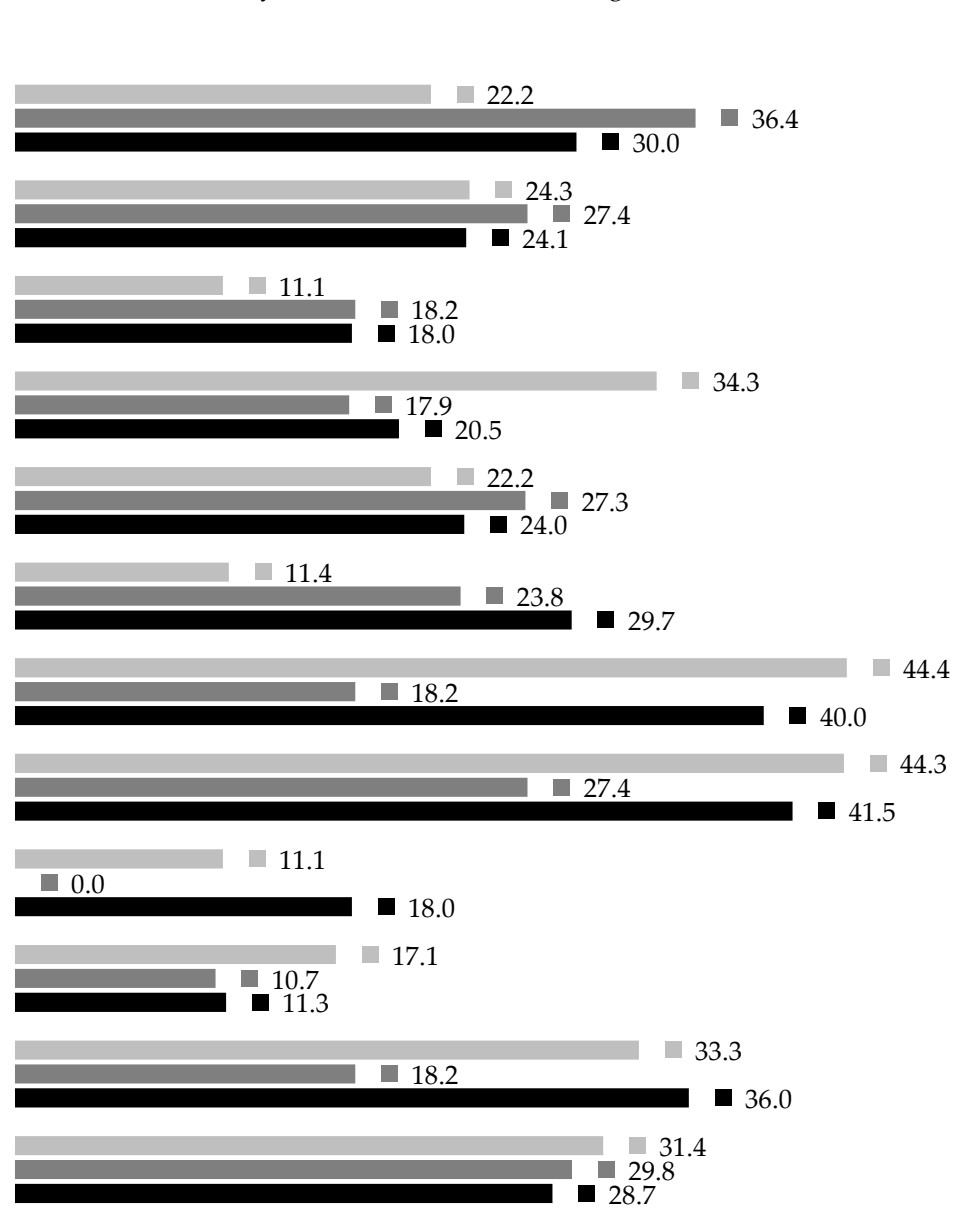

**Figure 3.** Distribution of the IBEX 35 companies Management Teams by position, gender, and training type. Source: SABI database. IBEX companies' annual reports.

In order to support or reject the hypotheses that have been formulated, we proceeded to perform an empirical analysis using the multiple regression model, to which predictors concerning the increase of the EBITDA (Table 3) and NET SALES (Table 4) have been added to verify its effect.

**Table 3.** Variables of influence on EBITDA in the IBEX 35 companies.

| Dependent Variable: EBITDA (Thousands) | | |
|---|---|---|
| *Multiple correlation (R) 0.610* | | |
| Independent variables | B standard | Student's T |
| GENDER (Men) | −0.197 | −2.310 * |
| AGE | | |
| AGE 1 (31–45) | 0.110 | 1.240 |
| AGE 2 (46–60) | 0.191 | 1.817 [#] |
| AGE 3 (61–70) | 0.297 | 2.844 ** |
| AGE 4 (+70) | 0.337 | 3.255 ** |
| NATIONALITY (Spanish) | −0.152 | −1.878 [#] |
| EDUCATION LEVEL | | |
| UNIVERSITY EDUCATION | 0.087 | 1.042 |
| DOCTORATE | 0.076 | 0.927 |
| DOUBLE DEGREE | −0.046 | −0.572 |
| ECONOMIC TRAINING | 0.142 | 0.848 |
| LEGAL TRAINING | −0.031 | −0.192 |
| TECHNICAL TRAINING | −0.043 | −0.293 |
| MASTER'S DEGREE | −0.082 | −0.907 |
| LANGUAGES | | |
| NATIVE LANGUAGE + 2 O + | 0.194 | 2.217 * |
| OTHER TRAINING | | |
| TRAINING SUITABLE FOR THE POSITION | −0.103 | −1.141 |
| FURTHER TRAINING | −0.397 | −4.515 *** |
| EXPERIENCE | | |
| EXPERIENCE OTHER SECTORS | 0.013 | 0.150 |
| INTERNATIONAL EXPERIENCE | −0.084 | −0.942 |
| $R^2$ 0.372 | | |
| F 3.487 *** | | |

Significance level: *** $p < 0.001$; ** $p < 0.01$; * $p < 0.05$; [#] $p < 0.1$; Own elaboration data source: SABI database. IBEX companies' annual reports.

**Table 4.** Influence variables in IBEX 35 companies NET SALES.

| Variable Dependent: NET SALES THOUSANDS | | |
|---|---|---|
| *Multiple correlation (R) 0.618* | | |
| Independent variables | B standard | Student's T |
| SEXO (Men) | −0.150 | −1.770 [#] |
| AGE | | |
| AGE 1 (31–45) | 0.175 | 1.980 * |
| AGE 2 (46–60) | 0.053 | 0.506 |
| AGE 3 (61–70) | 0.162 | 1.566 |
| AGE 4 (+70) | 0.201 | 1.960 [#] |
| NATIONALITY (Spanish) | −0.121 | −1.507 |
| EDUCATION LEVEL | | |
| UNIVERSITY EDUCATION | 0.022 | 0.265 |
| DOCTORATE | 0.134 | 1.659 [#] |
| DOUBLE DEGREE | 0.025 | 0.312 |
| ECONOMIC TRAINING | 0.102 | 0.614 |
| LEGAL TRAINING | −0.029 | −0.183 |
| TECHNICAL TRAINING | −0.100 | −0.693 |
| MASTER'S DEGREE | −0.082 | −0.906 |

**Table 4.** *Cont.*

| Variable Dependent: NET SALES THOUSANDS | | |
|---|---|---|
| *Multiple correlation (R) 0.618* | | |
| LANGUAGES | | |
| NATIVE LANGUAGE + 2 O + | 0.120 | 1.383 |
| OTHER TRAINING | | |
| TRAINING SUITABLE FOR THE POSITION | 0.215 | 2.384 * |
| FURTHER TRAINING | −0.456 | −5.226 *** |
| EXPERIENCE | | |
| EXPERIENCE OTHER SECTORS | 0.060 | 0.720 |
| INTERNATIONAL EXPERIENCE | −0.037 | −0.418 |
| $R^2$ 0.381 | | |
| F 3.632 *** | | |

Significance level: *** $p < 0.001$; $p < 0.01$; * $p < 0.05$; # $p < 0.1$; Own Elaboration Data source: SABI database. IBEX companies' annual reports.

According to Table 3, the multiple regression model explains a total of 37.2% ($R^2$) of the variance of the EBITDA result. Once the predictors are incorporated into the model, the demographic diversity variables, such as gender, nationality, and middle and older age brackets, are significant. Likewise, of the education diversity variables, only knowing two or more languages and complementary training are significant.

While we found that the Student's t-test of certain coefficients (age1, all training levels, training suitable for the position and experience levels) did not have statistical significance, the equation was, on the whole, significant at 0.001 (F = 3.487), which indicates that the model was generally valid.

If we analyze Table 4, with reference to the regression with the dependent variable, NET SALES of each IBEX 35 company, a value of $R^2$ is observed, which explains 38.1% of the variance of NET SALES. This explanation is reached by considering the significant variables (Student's t) obtained in the model, which, in this case, include gender, the youngest and oldest age groups, doctoral training level, suitable training for the position, and complimentary training.

The variables without statistical significance in this model include the rest of the variables shown in Table 4 (middle-age group, nationality, all training levels, except a doctorate, languages, experience in other sectors than the current one and international experience). The equation has a significance level F = 3, 632 of less than 0.001, and the model is, therefore, generally valid.

The non-statistical significance of the training variables could be due to the fact that, in principle, the work of a TMT is not of a practical nature, but one involving decision-making and creating a strategy to be deployed by the company. This supports the upper echelon theory, since the characteristics of each management team are unique to each company, as reflected in the results, which would explain that the doctorate training level variable that corresponds to theoretical training is significant for the NET SALES variable shown in Table 4. This same reason may explain the non-significance of the members' younger age group (Table 3), since in the creation of the company's organizational strategy, the members of the middle and older age group have a greater preponderance, which is mainly due, in our case, to the low proportion that this age bracket presents in the distribution (Figure 2).

To ensure that the estimates of all predictors are efficient, both variables that are individually significant and those that are not, when incorporated into the regression model, several tests have been carried out, all with significant results and confirming the explanatory capacity of the variables.

The endogeneity problem can be driven by unobservable managerial characteristics, as Coles and Li (2019a, 2019b) explain in their work. Li (2016) explored all the remedies for correcting endogeneity problems and found that GMM has the greatest correction effect on the bias. In our case, as it is not possible to apply this model because we do not have the panel data, the endogeneity has been resolved through the application of Hausman's test (Hausman 1978; Hausman and Taylor 1981).

*Hypotheses Contrast*

Based on the previous results, we contrasted the hypotheses formulated beforehand, a summary of which is shown in Table 5. Thus, H1a cannot be rejected, since the gender variable is significant, but in a negative sense, meaning in the absence of female members in a TMT, sales did not experience positive growth, which confirms that female diversity contributes to an increase in company sales (Smith et al. 2006; Carter et al. 2007).

Hypothesis H1b must be rejected because although members of the higher age bracket turned out to be significant (that is, their presence in the TMT favored an increase in sales), the younger age bracket had a higher level of significance. Incorporating younger TMT members can increase corporate profits, while older ones may have less physical and mental rigor and display risk aversion when making decisions, which may endanger financial security (Hambrick and Mason 1984; Wiersema and Bantel 1992).

We also cannot support hypothesis H1c, because the variable, NATIONALITY, is not significant and therefore does not influence the results of the companies analyzed. Some authors indicate that the introduction of TMT members of different nationalities into a company can sometimes be conducive to attracting potential foreign investors to the firm (Knežević et al. 2017). Moreover, hypothesis H1d cannot be fully supported, since it was formulated in a general sense, and not all the training variables were significant. We can confirm that, although no hypothesis specific to the concept was formulated, TMT members who hold a PhD or suitable training for the position have an influence in increasing sales, but it is not necessary to provide complementary training (significant negative variable). According to the Upper Echelons Theory, diversity in training can be related to the results of a business, which constitute a unique feature of each company (Ling and Kellermanns 2010; Minichilli et al. 2010).

**Table 5.** Comparison of the results of the hypotheses.

| Hypothesis | Result |
| --- | --- |
| **H1a:** The inclusion of female members in the IBEX 35 companies' management teams positively influences sales | *Supported* |
| **H1b:** The net sales of the IBEX 35 listed companies increase when members of the management team are older | *Rejected* |
| **H1c:** The inclusion of various nationalities in the management teams of the IBEX 35 companies favors an increase in sales | *Rejected* |
| **H1d:** The training of members of the IBEX 35 companies' management team contributes to an increase in the companies' sales | *Rejected* |
| **H2a:** The inclusion of female members in management teams has a positive effect on the EBITDA of the IBEX 35 companies | *Supported* |
| **H2b:** The final EBITDA of the IBEX 35 listed companies increases when the members of the management teams are older | *Supported* |
| **H2c:** The inclusion of various nationalities in the management teams of the IBEX 35 companies favors an increase in the final EBITDA | *Supported* |
| **H2d:** The training of the members of the IBEX 35 companies' management teams contributes to an increase in the companies' EBITDA | *Rejected* |

Regarding the formulated hypotheses related to the EBITDA dependent variable, H2a, H2b, and H2c are accepted. H2a is supported for the same reason as that given for H1a, i.e., the inclusion of female members in the IBEX 35 companies' management teams contributes to an increase in corporate EBITDA. H2b is supported because the upper age brackets are significant for EBITDA, that is, older TMTs favor an increase in corporate EBITDA, which could be related to their prudence when it comes to risking the companies' financial security (Hambrick and Mason 1984; Wiersema and Bantel 1992). As for H2c, we can confirm that in the absence of members of non-Spanish nationalities in the companies' management teams, the EBITDA was not increased, because, while it was a significant

variable, the result of the Nationality variable was negative. Therefore, the inclusion of different nationalities in the IBEX 35 companies' TMTs favors an increase in the final EBITDA. The diversity of nationalities in TMTs could serve to attract investment (Knežević et al. 2017).

Finally, H2d, like H1d, cannot be fully supported, since it was formulated in a general sense, and not all the training variables were significant. We can confirm that, although no specific hypothesis was formulated in this regard, TMT members who have training in two or more languages have an influence in increasing business EBITDA, but it is not necessary to provide complementary training (negative significant variable). So far, we have been able to confirm the explained theory, compare the hypotheses, and meet the objectives outlined in this study.

At this point, we are looking for new and relevant research channels related to corporate governance and social responsibility. In this sense, studies like that of Giroud and Mueller (2011), Li (2014, 2018), Coles et al. (2018), and Li et al. (2018) open up new perspectives on the analysis of, and future lines of research on, corporate governance and social responsibility. It should be borne in mind that our research objective, i.e., determining the effect of diversity in the senior management teams of listed companies, is a fundamental issue in strategic management and forms part of social responsibility policy. In the same line, studies on issues concerning compensation or incentives applicable to TMTs can be found in Core and Guay (1999), and literature on how CSR affects financial performance can be found in Li et al. (2019).

## 5. Conclusions

This study contributes to the expansion of the theoretical knowledge of the Upper Echelons Theory, considering the educational and demographic diversity of TMTs as a unique feature of companies, which, added to each member's skills, aids in making strategic decisions that contribute to the improvement of companies' performance, in our case, of the IBEX 35 companies.

From the results obtained, we can confirm, more specifically, that the inclusion of women in the management teams of this type of company has a positive influence on company sales and contributes to increasing the final EBITDA. Older TMT members favor an increase in the final EBITDA, but they do not guarantee an increase in sales. If management teams include members of different nationalities, the final EBITDA is increased, but company sales are not increased. Not all of the forms of training of the members influence the final performance, but we can confirm that having a doctorate level of training or suitable training for a position brings about an increase in sales. On the other hand, in order to achieve an increase in EBITDA, management team members must have knowledge of two or more languages, but companies do not need to provide complementary training to obtain this.

The analysis of the distribution of the members of the management teams of the IBEX 35 companies also provides interesting conclusions regarding the different proportions of both sexes, indicating a great imbalance in terms of the number of members (17% women/83% men in total). Thus, we can confirm a quantitative evolution experienced by female members over time, with respect to their age, the absence of such an evolution among the older members (70+ years), and a larger proportion in the middle age group, once the members of this group have received more training and, consequently, access to the labor market. This highlights the larger proportion of women who are managers and in dual positions with master's degrees and doctorate training levels, compared to men in the same positions, and this is repeated for female directors and managers in terms of knowledge of two or more languages. That is, the professional advancement of women in TMTs is linked to more training, compared to that of men.

From the point of view of business practice, the results obtained are useful for contributing to the knowledge of the characteristics of TMTs that are valid in order to obtain better results, when it comes to attracting or hiring the members of a management team and establishing responsible organizational policies that promote the inclusion of gender diversity. These results will, therefore, help to correct the imbalance between the number of women and men forming TMTs. In addition to promoting equality,

the incorporation of members of non-Spanish nationalities will also help each company to create a unique feature that would enrich it.

Despite the usefulness of the results obtained and their implications, this study presents some limitations that suggest future areas of research. In the first place, the data used in this study refer only to the IBEX 35 companies, excluding the banking sector, so the results are only applicable to the remaining 28 companies. The same type of research could be carried out in other business sectors, either to confirm the results obtained or, on the contrary, refute them. Likewise, it would be interesting to analyze other variables that would allow for more in-depth explanations of the dependent variables, such as those of a psychosocial nature that intervene in the strategic decision-making process and that can also enrich a company, although the collection of these data would require other methods than those used in this research, such as a survey of TMT members.

Finally, in future research, we must consider the relationship between the subject matter analyzed in this work, i.e., diversity in TMTs, with corporate governance and with the social responsibility policy implemented by listed companies.

**Author Contributions:** M.R.-F. designed the research framework and drafted the paper; A.I.G.-G. conducted the analysis, analyzed the data; E.M.S.-T. gave practical recommendations for the construction of the index system and edit the manuscript. All authors have read and agreed to the published version of the manuscript.

**Funding:** This research was partially funded by the 1st Research Plan of the Faculty of Social and Labour Studies of the University of Málaga.

**Conflicts of Interest:** The authors declare no conflict of interest.

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
