# Peer review of "Does Diversity in Top Management Teams Contribute to Organizational Performance? The Response of the IBEX 35 Companies"

_socsci, doi:10.3390/socsci9040036_

Round 1

Reviewer 1 Report

Dear authors.

Thank you for giving me the opportunity to review this paper.

You should review some aspects of this paper in order to gain readability and make it publishable.

First of all, MDPI journals follow a specific way of citing. You should adjust your work to that. Remember the references are written into [] instead of APA style, and in order of appearance.

The theoretical framework is too brief. Maybe you should develop it wider.In the hypotheses section it is not necessary to repeat them twice. Please just keep one form. I think you should introduce or at least mention your variables in the theoretical framework section. It would improve the readability of your paper.

Revise your tables and adjust them to MDPI format. You should check the use of the caps and the alignment of the text.

Best wishes on your publication.

Author Response

Dear Reviewer 1

Thanks a lot for the revision of our paper. We have performed the changes in the file submitted to the system.

We have activated the option “Track changes” with “Comments” on the right where you can see the responses to all your suggestions.

We hope the paper now comply with all your requirements.

With our best wishes,

The authors

Reviewer 2 Report

attached

Author Response

Dear Reviewer 2

Thanks a lot for the revision of our paper. We have performed the changes in the file submitted to the system.

We have activated the option “Track changes” with “Comments” on the right where you can see the responses to all your suggestions.

We hope the paper now comply with all your requirements.

With our best wishes,

The authors

Round 2

Reviewer 2 Report

Well done. Writing quality is still not satisfactory. Please (hire a professional editor to) proofread it.

Author Response

Dear Reviewer 2

Thanks a lot for the time and the effort in revising our paper. We have performed the changes in the file submitted to the system.

We have also performed an English editing service in order to improve the readability of the paper.

We have activated the option “Track changes” with “Comments” on the right where you can see the responses to all your suggestions.

We hope the paper now comply with all your requirements.

With our best wishes,

The authors